# Effect of Initial Predeformation on the Plastic Properties of Rolled Sheets of AISI 304L Austenitic Steel

**DOI:** 10.3390/ma15103575

**Published:** 2022-05-17

**Authors:** Jaroslaw Szusta, Aleksander Zubelewicz

**Affiliations:** 1Faculty of Mechanical Engineering, Bialystok University of Technology, 45C Wiejska Str, 15-351 Bialystok, Poland; 2Civil Engineering Department, University of New Mexico, Albuquerque, NM 87131, USA; alek.zubelewicz@gmail.com

**Keywords:** stress–strain measurements, thermal analysis, complex strain path, plastic anisotropy, reconfigurations of plastic flow, steel AISI 304L

## Abstract

This paper presents research on the influence of material anisotropy caused by the technological process of its manufacturing on the plastic properties of the material. In the experimental study, samples cut from an AISI 304L rolled sheet in the rolling direction, transverse, and at a 45° angle to the rolling direction were predeformed by axial deformation at 18 and 30%. The principal specimens extracted from the pre-deformed plates, cut in the longitudinal, transverse, and 45° angle directions, were subjected to tensile loading until failure. The data thus obtained allowed for the analysis of the plastic flow mechanism using the author’s calculation procedure. The *C*_R_ coefficient analysis provided information on the state of plastic anisotropy caused by the pre-deformation. For the specimens predeformed in the rolling direction, plastic flow isotropy was observed at a strain of 35%. For the specimens predeformed in the transverse direction—the plastic anisotropy is completely removed at a strain of 33%. For the specimens predeformed at 45 degrees to the rolling direction, it was found that the strain completely removed the plastic anisotropy induced by rolling. The calculations provided information that due to an abrupt change in the strain path, a strong reconfiguration of the plastic flow mechanism occurs, causing the removal of anisotropy generated by rolling.

## 1. Introduction

Material in the form of cold-rolled sheets shows anisotropy characterized by differences in plastic properties in three mutually perpendicular directions—in the direction perpendicular to the rolling direction, in the direction lying in the sheet plane, and the normal direction to the surface of the sheet (main directions of anisotropy). Sheet metal is widely used in many industries to produce, among other things, car bodies, cans, and cold-pressed machinery parts. Machine parts are often manufactured by cold stamping. The process minimizes the number of operations to reduce the costs of manufacturing. The aim is to obtain a flawless product with a smooth surface on which protective coatings can be applied directly. In this process, in addition to the friction force, the anisotropy of the material has a significant effect on the distribution and value of strain and thus on the quality of the product. It is sufficient to “form” the plastic properties of the material so that the molding process takes place in a controlled manner. The control of plastic properties of a material can be carried out, for example, by using predeformation loading in desired directions [1]. The crystallographic texture of cold-rolled and annealed sheets strongly depends on the technological process parameters of their manufacture. This process determines the anisotropy of the plastic properties of the material [2,3].

Very often, the crystallographic texture of a material is not homogeneous—it changes as a function position within the sample. For example, in rolled samples, the texture is strongly dependent on the depth from the surface into the sample. Accurate knowledge of texture is essential for a variety of applications most commonly related to the prediction of macroscopic material properties based on the anisotropy of a given property at the monocrystalline level [4,5,6]. The crystallographic texture of the sheets affects the strain distribution and plastic flow during the moulding process of the products. Therefore, it is plastically deformed to the appropriate level to obtain a material with a specific anisotropy. In rolled sheets of metals and alloys, plastic anisotropy results from the crystallographic alignment of slip systems in an otherwise polycrystalline material. The large plastic deformations caused by the rolling process results in reorientation and fragmentation of the grains. This has been documented in many experimental and theoretical works [1,2,3,4].

These studies have led to the development of various anisotropy coefficients where the Lankford coefficient *R* and its derivatives provide a quantitative measure that is used by both researchers and engineers [5]. Typically, the coefficients are calibrated under proportional loading conditions, where the textured material is tested along the rolling direction, transverse direction, and at 45 degrees to the rolling direction. The *R*-coefficients are specified at sufficiently large deformation, thus allowing for accurate measurement of the lateral plastic strains.

Anisotropic yield surfaces are experimentally evaluated in the condition of a constant strain rate. In textured materials, the uniaxial tests in tension are often conducted in different orientations with respect to the rolling direction. In addition, biaxial tests in tension at various stress ratios are used to determine the points on the stress loci [6,7,8,9,10]. It is worth noting that plastic hardening somewhat complicates the analysis, where it is not entirely clear how to choose the proper amount of plastic work or strain range at the given hardening rate. The measurements are also sensitive to the deformation protocols, making a significant difference in whether the experiment is conducted under proportional or non-proportional loading. For these reasons, the forming limit diagram constructed in terms of strains is considered a viable indicator of plastic anisotropy [11,12]. The authors of the work [13,14,15,16] used this indicator to analyse the behaviour of plastically anisotropic materials subjected to loads. Among the numerical methods, the self-consistent anisotropic approach was recognised [17] and proven to correctly predict the anisotropic behaviour of various polycrystalline materials.

Metals and alloys exhibit complex behaviour under non-proportional loading. It has been reported that an abrupt change of the strain path triggers equally strong reconfiguration of the plastic flow [18,19]. Furthermore, the materials subjected to an orthogonal strain path tend to form micro-shear bands [20].

The crush-forming of stainless steels (AISI 304L) as a result of cold plastic deformation caused by the technological process of rolling creates high strength properties [21,22,23,24]. The phenomenon of strengthening by crushing distinguishes these steels from most other materials. Stainless austenitic and ferritic austenitic steels, when plastic deformed, show an interesting combination of high strength and formability, making it possible to reduce the weight of the parts. The mechanical properties of AISI 304L can be improved if the stainless steel is cold-formed before basic forming [25,26,27,28]. This property makes the material well suited for the production of roll-formed profiles to reduce the weight of the structures they are used in. These profiles, made from a pre-formed AISI 304L steel strip, are used for car bodies, train bodies, and various structural frames. Lighter structures use less energy during acceleration and deceleration, which is an extremely important economic aspect. In responsible mechanical structures, structural isotropy of the material is aimed for, which is problematic in the case of prefabricated products manufactured by cold forming technology [29,30,31].

The problem discussed in this paper is very important from a technological point of view. Many papers are concerned with the analysis of the mechanical and corrosion properties of stainless steel produced using different technologies. The effects of various influences on the final properties of the material are studied, including pre-deformation [32,33,34]. However, there is no unambiguous answer as to how the plastic properties of the material should be formed to obtain their required directional distribution.

For parts manufactured from thin cold-rolled sheets, e.g., by pressing, isotropy of plastic properties is often required. One of the ways to achieve this effect is a directional predeformation of the material. This paper presents a method for determining the change of material anisotropy as a function of strain level. This study aims to measure, then theoretically evaluate, the plastic flow reconfigurations in steel AISI 304L, where the reconfigurations are triggered by the changing pathways of the tensile strain. Austenitic chromium-nickel steel is widely used in the petrochemical, chemical, food, automotive, and other industries, where significant structural components are made from it. Therefore, it is important to know how to pre-treat the material and how to orient the cut pieces on the sheet metal; thus, their properties are the same in all directions.

To verify the study aims, 21 configurations of test specimens were prepared and subjected to uniaxial tension until failure. This provided the data needed to develop a method for determining the variation in plastic properties of the material. Figure 1 shows the essence of the work in which the parameter that determines the plastic properties of the material is the *C*_R_ coefficient. Details of the experimental studies are included in Appendix A of the paper.

## 2. Influence of The Direction of Initial Predeformation on the Plastic Anisotropy

### AISI 304L Steel

Often, plastic anisotropy is monitored in terms of the Lankford coefficient, where the ratio of two lateral plastic strains is taken at a sufficiently large magnitude of the axial strain. The coefficient is regularly incorporated into the descriptions of the anisotropic yield surfaces. In this approach, it is moved one step back, and instead, a new coefficient of anisotropy is derived. The analysis suggests that a rate-based coefficient more correctly captures the instantaneous character of the plastic flow. In order to make the study tractable, the experimental data obtained from all test protocols were converted into appropriate polynomials. More specifically, polynomials were constructed where the true plastic strains and the true stress are expressed in terms of the true axial strain. This ensures that the error of the approximations is negligibly small. In each test protocol, the loading pathway was specified in terms of the deformation path accumulated during proportional and/or non-proportional loading. In other words, the deformation path is treated similarly as if it was the adequate time of the active loading process.

The analysis starts with selecting the mechanisms of plastic flow that were justifiable by physics. It is noted that the general rule of the Huber–Mises mechanism can be applied to stainless steel. In this approach, the mechanism is constructed in the framework of tensor representation [35,36]. As a result, the plastic flow tensor is introduced $\;M=M^{\sigma }+C_{1}^{R}\;\left(N^{a}-N^{w}\right)+C_{2}^{R}\left(N^{a}-N^{z}\right)+C_{3}^{R}\left(N^{w}-N^{z}\right)$ such that:(1)$${\dot{\varepsilon }}^{p}=\frac{1}{2}M\;{\dot{e}}_{eq}$$

In the Cartesian coordinate system, the symmetric plastic flow tensor $\;M_{ij}$ has six components, where  i,j=1,2,3. The tensor consists of the isotropic Huber–Mises mechanism $\;M^{\sigma }=\sqrt{3}\;S/\sqrt{J_{2}}$ and there are three slip orientations aligned with the active plastic processes. In this notation, the stress deviator is  S=σ−1 trσ/3 and the true stress is  σ. The second invariant of the stress deviator is $\;J_{2}=S:S/2$, where the double colon indicates that it is a scalar product of the tensors. Herein, the directional plastic flow may occur along three planes of B samples, namely the plastic flow on the plane  {a,w} is defined by the tensors $\;\left(N^{a}-N^{w}\right)$, the plastic flow on  {a,z} is $\;\left(N^{a}-N^{z}\right)$ and, lastly, the plastic flow on  {w,z} is $\;\left(N^{w}-N^{z}\right)$. The three tensors $\;N^{a}$, $N^{w}$ and $\;N^{z}$ are dyadic products constructed on the basis of unit vectors $\;n^{a}$, $n^{w}$ and $\;n^{z}$. The vectors are pointing in the direction of the active loading $\;n^{a}$, and the later direction $\;n^{w}$ and the through-thickness direction $\;n^{z}$. In this manner, the tensors are $\;N^{a}=n^{a}⊗n^{a}$, $N^{w}=n^{w}⊗n^{w}$ and $\;N^{z}=n^{z}⊗n^{z}$, respectively. The unit vectors are orthogonal and, therefore $\;1=N^{a}+N^{w}+N^{z}$, where the identity tensor is **1**. The polynomials of the stress–strain curves and elastic properties are used for the prediction of the plastic strain rates. In the next step, the experimentally obtained strain rates were fed into the rates specified by Equation (1). As a result, three rates of plastic strain were defined, where the first is aligned with the axial deformation $\;{\dot{\varepsilon }}_{a}^{p}=N^{a}:{\dot{\varepsilon }}^{p}$. The through-width strain rate is $\;{\dot{\varepsilon }}_{w}^{p}=N^{w}:{\dot{\varepsilon }}^{p}$ and the through-thickness strain rate becomes $\;{\dot{\varepsilon }}_{z}^{p}=N^{z}:{\dot{\varepsilon }}^{p}$. It is assumed that the material is plastically incompressible  (trM=0) and, therefore, we have $\;{\dot{\varepsilon }}_{z}^{p}=-\left({\dot{\varepsilon }}_{w}^{p}+{\dot{\varepsilon }}_{a}^{p}\right)$. In the analysis, the current stress is $\;\sigma =N^{a}\sigma_{a}$ and, therefore, the plastic flow tensor becomes $\;M=\left(3N^{a}-1\right)+C_{1}\;\left(N^{a}-N^{w}\right)+C_{2}\left(N^{a}-N^{z}\right)+C_{3}\left(N^{w}-N^{z}\right)$. Consequently, the measured rates of plastic strain $\;{\dot{\varepsilon }}_{a}^{p}$ and $\;{\dot{\varepsilon }}_{w}^{p}$ are:(2)$$\begin{array}{c} {\dot{\varepsilon }}_{a}^{p}=\left(1+C_{1}/2+C_{2}/2\right){\dot{e}}_{\mathrm{eq}}\\ {\dot{\varepsilon }}_{w}^{p}=-\left(1/2+C_{1}/2-C_{3}/2\right){\dot{e}}_{\mathrm{eq}}\\ {\dot{\varepsilon }}_{z}^{p}=-\left(1/2-C_{2}/2-C_{3}/2\right){\dot{e}}_{\mathrm{eq}} \end{array}$$
where the through-thickness rate is calculated. In plastically isotropic material, it must be ensured that the three coefficients $\;C_{1}$, $\;C_{2}$ and $\;C_{3}$ are equal to zero. The equations are solved, and it was noticed that the three coefficients were reduced to just one:(3)$$C_{R}=1+2\;\frac{{\dot{\varepsilon }}_{w}^{p}}{{\dot{\varepsilon }}_{a}^{p}}$$
while the equivalent plastic strain becomes $\;{\dot{e}}_{eq}={\dot{\varepsilon }}_{a}^{p}$. The isotropic plastic flow requires that $\;{\dot{\varepsilon }}_{w}^{p}=-{\dot{\varepsilon }}_{a}^{p}/2$, and then the coefficient $C_{R}$ is equal to zero. In its final form, the plastic flow tensor becomes:(4)$$M=\frac{\sqrt{3}\;S}{\sqrt{J_{2}}}+C_{R}\;\left(N^{w}-N^{z}\right)$$

In the next step, the equivalent stress is derived. It must be ensured that the requirement of plastic work invariance [35] is always satisfied. This requirement states that the rate of plastic work is independent of the frame of description, hence $\;\sigma :{\dot{\varepsilon }}^{p}=\sigma_{eq}{\dot{e}}_{eq}$:(5)$$\sigma_{\mathrm{eq}}=\frac{1}{2}\;M:\sigma$$

Note that the rate of plastic strain is defined in (1), and the plastic flow tensor is given in (4). In this construction, the equivalent stress is described in terms of the Huber–Mises plastic flow mechanism $M^{\sigma }$ and is a function of the dynamic coefficient of anisotropy $\;C_{R}$. The coefficient quantifies the directionality of plastic flow and explicitly affects the shape of the yield surface, where the yield surface is $\;\sigma_{eq}=\sigma_{a}$. The stress contours are determined in terms of the current stress $\;\sigma_{a}$. This stress is rescaled by an arbitrarily chosen reference stress $\;\sigma_{a}^{0}$. Accordingly, we choose the axial stress that is measured in A1B1_0 sample at strain 0.9%. In summary, the following were obtained: stresses, the rates of plastic strain, and the coefficient *C_R_*—all the quantities are determined at each point of the deformation path. The path is the cumulative axial strain achieved during the proportional and non-proportional test protocols. The plastic strain rates are not quite constant, but their variations are sufficiently small throughout the entire deformation process such that these variations do not impact the stress, the plastic flow mechanisms, and the dynamic coefficient *C_R_*.

## 3. Results

As mentioned earlier, the test protocols are summarized in the Table A2 of Appendix A. The research assumed the adoption of 21 test protocols, where each test was repeated three times. First, the three B samples obtained from the virgin (non-deformed) A1, A2 and A3 samples were tested. The second group of the A1, A2, and A3 samples was pre-deformed to 18% of true strain. Each such sample was unloaded and then used to extract B1, B2, and B3 samples. The last group of large samples was pre-deformed to 30% of strain and after unloading, the final group of the B samples was obtained. In our notation, A1B1_0 means that the B sample was obtained from the non-deformed A1 sample. Consequently, the B sample was subject to loading in the B1 direction until failure. The label A3B2_18 indicates that the A3 sample was initially pre-deformed to 18% of the true strain, then the B sample was deformed in the B2 direction until failure. This notation was consistently applied to all samples. The stress–strain responses in all B samples were measured (Figure 2).

There are interesting trends worth noting. An increased rate of plastic hardening was observed in the A1B1_18, A2B1_18, and A3B1_18 samples. All the samples were preloaded, unloaded, and reloaded in the same direction. It has been argued that the increase in the hardening rate may result from the twin-slip interactions [20,21,23,33]. The non-proportional loading protocols consistently caused a noticeable stress overshoot. Clearly, the pre-existing dislocation structures imposed additional constraints on the cross-slip mechanism. The ultimate tensile stress (UTS) under all loading protocols was in the same range, and its mean magnitude was 1134 MPa, with the standard deviation being 36 MPa. The mean ductility was 48 percent, and the standard deviation was 2.6 percent.

Due to the technological process of its manufacture, the tested material had a higher dislocation density and thus higher internal energy than the non-deformed metal [37,38]. This results in the occurrence of anisotropy of mechanical properties. Directional plastic deformation (predeformation) causes elongation of individual grains in the flow direction of the material. The elongation of grains is accompanied by the ordering of their crystallographic axes, characterized by the parallelism of specific planes and crystallographic directions of individual grains. The directed deformation of the grains thus causes a change in mechanical properties depending on the flow direction of the material. In [39], the authors considered using similar AISI316L steel for hip acetabular cups. However, despite its good biomedical properties, this material, due to the anisotropy of its mechanical properties, proved to be worse at reducing Tresca stresses than it was considered by others. Predeformation of the material introducing targeted grain crosslinking could be an effective method to counteract this, which was confirmed in this work.

A summary of this research is grouped and presented in Figure 3, Figure 4 and Figure 5. The first group consists of the material responses on B samples obtained from A1 samples (Figure 3). The next group represents the predictions made on B samples extracted from A2 samples (Figure 4). The last set of results is gathered from B samples based on A3 samples (Figure 5). In the first column of each figure, the dynamic coefficients *C_R_* as they evolve during the active deformation path were displayed. For convenience, the loading direction is shown in the sketch of the A sample. The three contours $\;\sigma_{eq}=\sigma_{a}$ of the yield stress are constructed on the planes of principal stresses  (a,w), (a,z) and  (w,z). As mentioned earlier, the yield stress $\;\sigma_{a}$ is normalized by the axial stress (A1B1_0) taken at strain 0.9%. This stress is $\;\sigma_{a}^{0}$ = 240 MPa. The contours are defined at three deformation points, namely at strains equal to 19% (point 1), 31% (point 2), and 40% (point 3). Colour-coding has been applied here. The anisotropy coefficient *C_R_* and stress contours marked in black, represent the uninterrupted proportional tests (baseline tests). The blue lines depict results obtained on samples pre-deformed to 18% of true strain. The red lines refer to the samples preloaded to 30% of true strain.

### 3.1. Pre-Deformation in the Rolling Direction

In the A1B1_18 sample (blue lines in Figure 3), the coefficient *C*_R_ only slightly deviated from the baseline coefficient (black lines). However, the plastic flow mechanism in the A1B1_30 sample experienced an abrupt change. It was noticed that the stress contours in points 2 and 3 are not much different from each other. The dynamic coefficient was plotted as a function of the deformation path. The contours of the yield stress were constructed on three planes of principal stresses at deformation points 1, 2, and 3.

This was not a surprise because the coefficients *C_R_* in these points have similar values. The non-proportional loading brings the coefficient *C_R_* to zero, where the plastic flow becomes isotropic. At strains near failure, the coefficient *C_R_* increased again. It was observed that this pattern repeated itself in nearly all samples subjected to the complex loading protocols (Figure 3, Figure 4 and Figure 5). The behaviour was quite different in the reloaded samples at 45 degrees to the rolling direction.

In the A1B2_18 and A1B2_30 samples, the plastic flow mechanisms did not follow the baseline pattern (A1B1_0 sample). The abrupt reconfiguration of the plastic flow mechanism was also detected on the stress contours $\;\left(\sigma_{a},\sigma_{w}\right)$, $\left(\sigma_{a},\sigma_{z}\right)$ and $\left(\sigma_{w},\sigma_{z}\right)$. At strain 35%, the A1B2_18 sample exhibited nearly isotropic plastic flow. In this case, the plastic deformation removed the rolling-induced anisotropy.

In the A1B3_18 sample, the coefficient *C_R_* takes values close to zero, and as a result, the plastic flow becomes nearly isotropic. However, large deformations are responsible for the re-development of a directional plastic flow. Small differences in the coefficient *C_R_* made the stress contours closely spaced.

### 3.2. Pre-Deformation in the Transverse Direction

The coefficient *C_R_* indicates that this test protocol triggered an abrupt reconfiguration of the plastic flow mechanism. The baseline coefficient *C_R_* was re-established as the deformation continued, but the near-failure strain reintroduced the directional plastic flow.

The behaviour in the A2B2_18 and A1B2_18 samples were very similar. The plastic anisotropy is completely erased at strain 33%. It should be emphasized that the baseline anisotropy remained almost unchanged. The differences in the coefficient *C_R_* are clearly displayed on the stress contours.

This test protocol produces conditions suitable for erasing the rolling-induced anisotropy. Consistent with the other test results, the near-failure plastic flow became anisotropic again.

### 3.3. Pre-Deformed Sample at 45 Degrees to the Rolling Direction

In all A3B1 test protocols, the applied deformation erased the rolling-induced plastic anisotropy. The trend is seen in the plots of the coefficient *C_R_*. Consequently, the stress contours remained nearly symmetric with respect to the stress axes.

The strongest anisotropy was detected in the A3B2_30 sample. The plastic flow mechanism in the A3B2_30 sample evolved, causing a reduction of the anisotropy of the subsequent deformation. Anomalous behaviour was observed in the A3B2_18 sample, where the mechanism departed from the baseline mechanism, then the plastic flow mechanism became increasingly anisotropic.

As in all other cases, the near-failure strain reversed the trend. Note that the stress contours $\left(\sigma_{a},\sigma_{w}\right)$ and $\left(\sigma_{w},\sigma_{z}\right)$ consistently shifted toward the stress $\;\sigma_{w}$. At the same time, the contours on the plane $\left(\sigma_{a},\sigma_{z}\right)$ remained nearly symmetric. In summary, the A3B2 test protocol produces very different responses than the other tests.

An abrupt reconfiguration of the plastic flow mechanism is observed in the A3B3_18 sample. It should be emphasized again that plastic deformation tends to reduce the initial anisotropy.

## 4. Conclusions

The investigations carried out in this work prove that an abrupt change of the strain path induces a reconfiguration of the plastic flow mechanisms, as shown in the dynamic *C*_R_ coefficient waveforms. Moreover, the complex strain paths reduce, and in some cases eliminate, the anisotropy caused by the technological process of rolling the analysed AISI 304L sheet. The dynamic *C*_R_ factor presents the plastic flow mechanism at a given moment of loading. Due to this property, changes in the anisotropy of the material were observed for the tested specimen configurations and loads. It was shown that for load levels close to the failure of the AISI 304L material, the strain-reconfigured plastic flow mechanism became anisotropic again in some cases. The level of elimination of the anisotropy of the plastic properties of the material cannot be unambiguously determined for the load configurations analysed. Each case must be considered individually on the basis of the courses of the dynamic *C*_R_ coefficient. For the analysed cold-rolled AISI304L sheet, it is possible to generalize:For the pre-deformation in the rolling direction:

An increase in deformation induced by non-proportional loading results in a *C*_R_ value close to zero, which is evidence that the material has acquired isotropic plastic properties. Further deformation induces an anisotropy of its properties in the material again but less than the original one.

For the pre-deformation in the transverse direction:

For the A2Bx specimens cut in the direction perpendicular to the rolling, a reconfiguration of the plastic flow mechanism is also observed during the loading process. The change in the anisotropy of the material is more pronounced the higher the strain level.

For the pre-deformed sample at 45 degrees to the rolling direction

For all strain configurations of the A3Bx specimens, there was complete removal of the anisotropy of the mechanical properties of the cold-rolled AISI304L sheet already at a strain level of 38%.

The approach presented in this paper to determine the degree of anisotropy of plastic properties of directionally deformed material was verified for cold-rolled AISI 304L sheet. If the method is to be used for forms other than cold-rolled thin sheets, further verification tests should be carried out, which will indicate the limitations of the presented approach. This will be the subject of forthcoming papers.

## Figures and Tables

**Figure 1 materials-15-03575-f001:**
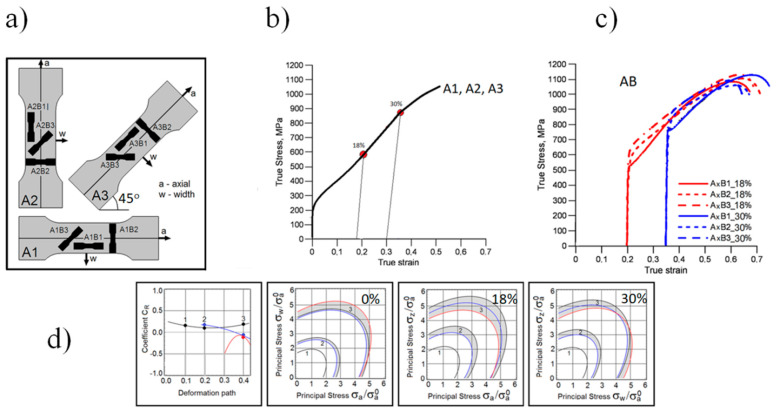
Graphic abstract of work: (**a**) orientation of test specimens on a cold-rolled sheet; (**b**) predeformation of type-A specimens—18 and 30%; (**c**) monotonic tensile test of type-B specimens cut from predeformed type-A specimens; (**d**) the course of the *C*_R_ coefficient determining plastic properties of the material.

**Figure 2 materials-15-03575-f002:**
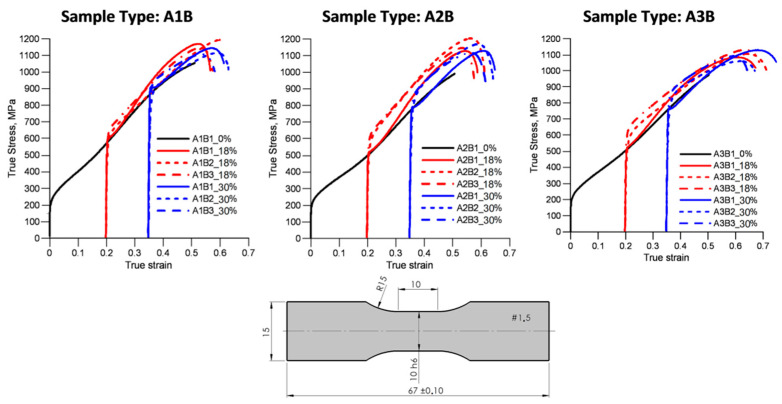
Stress–strain measurements in samples subjected to tensile deformation under 21 test protocols. Stress overshoot is clearly observed in the samples subjected to non-proportional strain paths. An increase in the plastic-hardening rate is depicted in the samples loaded and reloaded in the same direction.

**Figure 3 materials-15-03575-f003:**
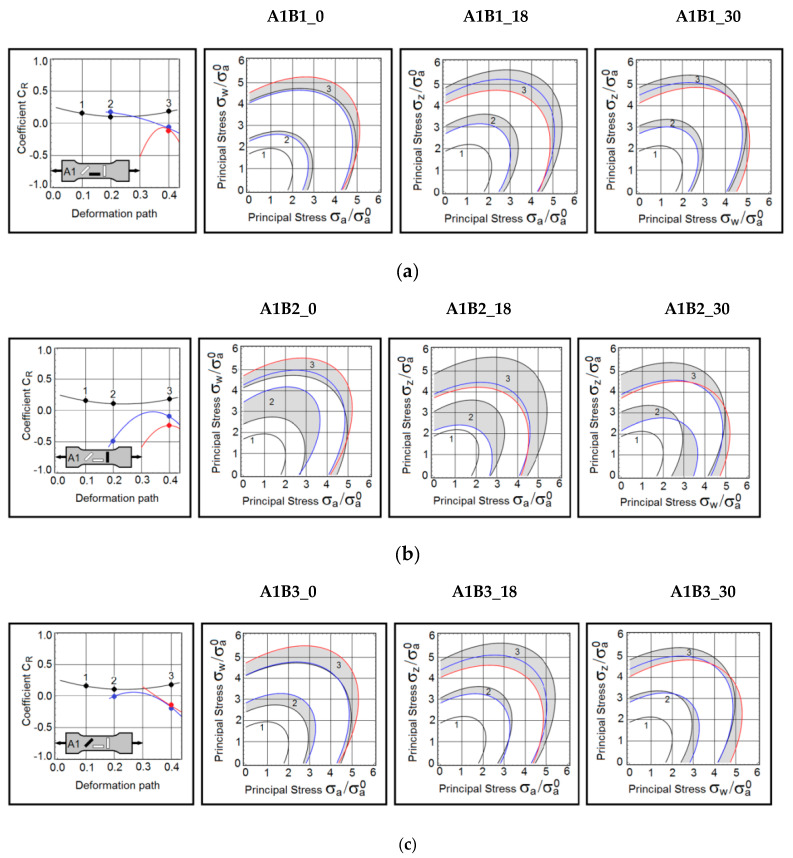
(**a**) Dynamic coefficient of anisotropy *C_R_* and stress contours in B samples extracted from A1 samples—A1B1 samples reloaded in the rolling direction. (**b**) Dynamic coefficient of anisotropy *C_R_* and stress contours in B samples extracted from A1 samples—A1B2 samples reloaded in the transverse direction. (**c**) Dynamic coefficient of anisotropy *C_R_* and stress contours in the B samples extracted from A1 samples—A1B3 samples reloaded at 45 degrees to the rolling direction.

**Figure 4 materials-15-03575-f004:**
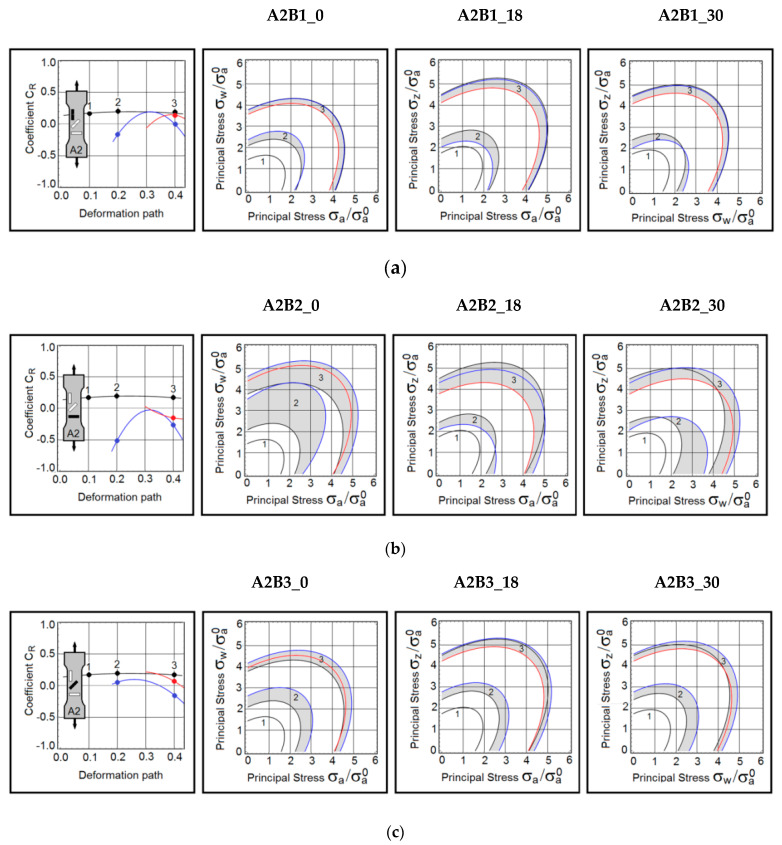
(**a**) Dynamic coefficient of anisotropy *C_R_* and stress contours in B samples extracted from A2 samples—A2B1 samples reloaded in the transverse direction. *(***b**) Dynamic coefficient of anisotropy *C_R_* and stress contours in B samples extracted from A2 samples—Samples A2B2 reloaded in the rolling direction. (**c**) Dynamic coefficient of anisotropy *C_R_* and stress contours in B samples extracted from A2 samples—Samples A2B3 reloaded at 45 degrees to the rolling direction.

**Figure 5 materials-15-03575-f005:**
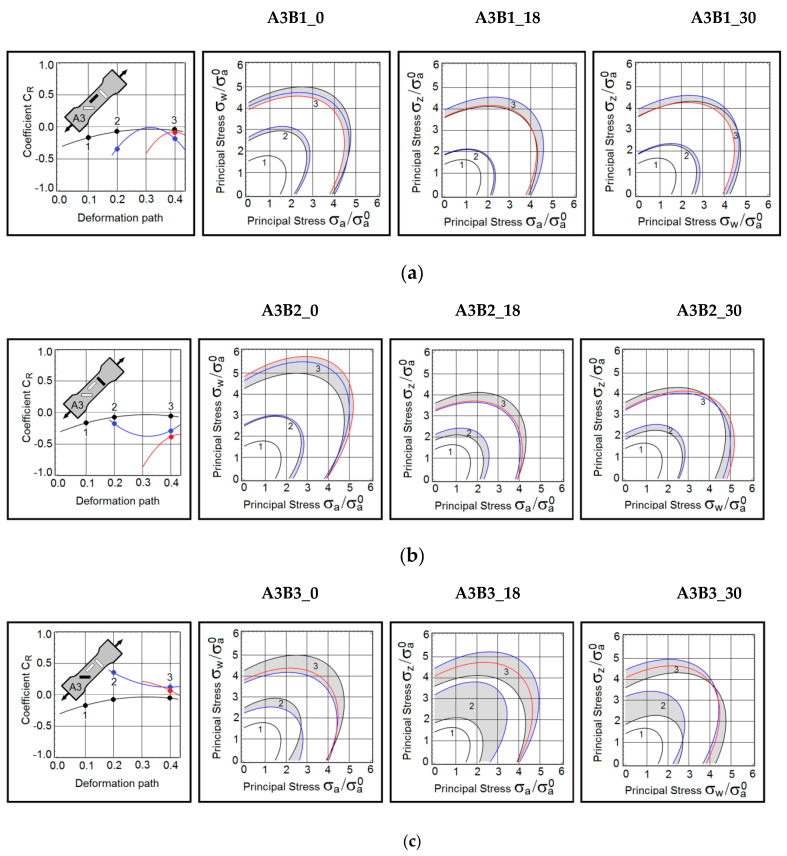
(**a**) Dynamic coefficient of anisotropy *C_R_* and stress contours in B samples extracted from A3 samples—A3B1 samples reloaded at 45 degrees to the rolling direction. *(***b**) Dynamic coefficient of anisotropy *C_R_* and stress contours in B samples extracted from A3 samples—A3B2 samples reloaded at 45 degrees to the transverse direction. (**c**) Dynamic coefficient of anisotropy *C_R_* and stress contours in B samples extracted from A3 samples—Samples A3B3 reloaded in the rolling direction.

## Data Availability

Experimental data obtained during the study, can be accessed from the corresponding author.

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
