# Peer review of "Effect of Initial Predeformation on the Plastic Properties of Rolled Sheets of AISI 304L Austenitic Steel"

_materials, 2022, doi:10.3390/ma15103575_

Round 1

Reviewer 1 Report

Reviewer comments: The following points needs to be considered/addressed before the manuscript can be accepted for publication.

  1. The Abstract is not written properly. This section looks more like an Experimental/Methods, rather than Abstract. The authors need to address/discuss their results and conclusion out of it.
  2. I feel like the way that the Introduction section written has no shape. There is some confusion in understanding of the information provided in this section; particularly I notice the jumbling of few sentences in the first paragraph of Introductions section. The authors need to go through this whole section and make it in an order. This section has to be in a pyramid shape with broader ideas at the bottom and ongoing above to become tip and narrow. But it does not seem to be like that.
  3. There are so many grammatical errors in the whole manuscript. The authors need to check and rectify the English mistakes. One example is, the word “the” has been used too many times, please check.
  4. The references provided are too old and not recent. The authors need to provide the references during the last 5-6 years so as to infer that the problem they selected is not outdated ones. I see there are references more than 50 years old.
  5. Please check the statement, “The crystallographic texture is an extremely important parameter of the cold-rolled and annealed steel strip. If the grain volume of every possible crystal orientation is the same in steel, the crystallographic texture is random and it is very difficult to obtain it.” It is confusing.
  6. Please check, “In rolled sheets of metals and alloys, plastic anisotropy results from the crystallographic alignment of slip systems in an otherwise polycrystalline material”.
  7. The statement, “Decades of experimental studies and theoretical developments provide a commonly-accepted interpretations of the behavior”, what kind of behavior that the experiments in the last decades provide?
  8. There is no Rationale for the work done in the Introduction section, please provide.
  9. Regarding the sample materials, it will be good to the readers if the authors can incorporate some physical characterization like XPS, EDX, porosity etc.
  10. Some cumulative discussion of results at the end of Results needs to be provided. The authors have provided only the information about their results obtained. However, the cumulative analysis of their obtained results with respect to a broader aspect is missing.

Author Response

Thank you again for your response concerning our manuscript entitled “Effect of initial predeformation on the plastic properties of rolled sheets of AISI 304L austenitic steel”. I have revised the manuscript according to reviewers’ comments. In the text marked in red made all the changes suggested by the reviewers. In addition, the structure of the article was restructured by including information about the experimental studies in the appendix. The table below contains the answers to reviewers suggestions. I hope the changes are sufficient to meet the standards of Materials both in terms of English language and manuscript content. Thank you for your time.

The abstract has been rewritten.

The introduction has been modified and expanded to include more recent literature.

Questionable sentences have been rewritten.

Microstructural studies of the deformed material are provided in the Appendix.

The summary has been rewritten.

Reviewer 2 Report

Dear Szusta and Zubelewicz,

The manuscript “Effect of initial predeformation on the plastic properties of rolled sheets of AISI 304L austenitic steel” (materials-1688874) by Szusta and Zubelewicz et al. studied the phenomena of plastic anisotropy in polycrystalline materials results from the mutual alignment of slip systems in textured steel AISI 304L subjected to tensile non-proportional strain paths. The topic is interesting, but I think this article should reconsider after proper changes in major revision for publication in Materials. Some of my specific comments are:

  1. The authors need to include quantitative results rather than only qualitative results in the abstract section.
  2. Describe the novelty of the article made by the author? From the results of my evaluation, it seems that many similar published works adequately explain what you have raised in the current manuscript. As the best reviewer knowledge in this research area, plastic properties in metal materials, especially AISI 304L have been widely studied in the past time in the perspective of analytical, experimental, and computational. If there is something others really new in this manuscript, please highlight it more clearly in the introduction section.
  3. The state of the art and the significance of the current study are not clearly present, the authors should highlight it more advanced in the introduction section.
  4. In the introduction section, the authors should explain the previous research conducted and its shortcomings. It will uphold the research gap that you filled with your research novelty. I recommend the authors elaborate their introduction section. Do not forget to attention carefully to my previous comments on numbers 1 and 2.
  5. Since this present study is related to the performance of metals, I would encourage and advise the authors to adopt some of the specific additional references related to metals evaluation in the introduction section published by MDPI as follow:
    • Tresca Stress Simulation of Metal-on-Metal Total Hip Arthroplasty during Normal Walking Activity. Materials (Basel). 2021, 14, 7554. https://doi.org/10.3390/ma14247554
  1. In materials and methods section, the authors should add one systematic figure to illustrate the workflow of the present study to make the reader more interested and easier to understand rather than only using dominant text and specific figures to explain.
  2. What is the software used to present Figure 5? It should be mentioned.
  3. The authors should explain the basis of specimen used and experimental performed.
  4. The authors must provide a detailed specification and use condition more detail regarding all tools used in the research carried out so that the reader can estimate the accuracy and differences in the results that the authors describe due to the use of different tools in future studies.
  5. The authors are advised to compare the results they obtain with previous similar/identical studies if it is possible.
  6. In the last paragraph before Conclusions, the authors should add one paragraph about the limitations of the research conducted.
  7. The conclusion of the present manuscript is not solid, further elaboration is needed.
  8. Further research needs to be explained in the conclusion section.
  9. In the whole of the manuscript, the authors sometimes made a paragraph only consisting of one or two sentences that made the explanation not clearly understood. The authors need to extend their explanation to become a more comprehensive paragraph.
  10. I see some error in English used by the authors. To improve the quality of English used in this manuscript and make sure English language, grammar, punctuation, spelling, and overall style are correct, further proofreading is needed. As an alternative, the authors can use the MDPI English proofreading service for this issue.
  11. All of the typesetting is not suitable with Materials, MDPI format correctly. The authors can download published manuscripts by Materials, MDPI, and compare them with the present author's manuscript to ensure typesetting is appropriate.

I am pleased to have been able to review the author's present manuscript. Hopefully, the author can revise the current manuscript as well as possible so that it becomes even better. Good luck for the author's work and effort.

Best regards,

The Reviewer

Author Response

Thank you again for your response concerning our manuscript entitled “Effect of initial predeformation on the plastic properties of rolled sheets of AISI 304L austenitic steel”. I have revised the manuscript according to reviewers’ comments. In the text marked in red made all the changes suggested by the reviewers. In addition, the structure of the article was restructured by including information about the experimental studies in the appendix. The table below contains the answers to reviewers suggestions. I hope the changes are sufficient to meet the standards of Materials both in terms of English language and manuscript content. Thank you for your time.

The abstract has been rewritten.

The introduction was modified and stated the purpose of the research conducted.

The literature has been completed and updated.

The introduction provides a graphic overview of the methodology of the study.

Information on the software used has been completed.

The limitations of using the presented approach are defined in the conclusion of the paper.

The summary has been rewritten.

Reviewer 3 Report

The paper is written poorly with too many grammatical errors. I only reviewed the abstract and the intro parts for now, and already found too many errors that prevent me from obtaining what the authors are trying to say. Without going much into the technical details, I have to recommend against the publication of this paper. I would suggest a thorough revision throughout the manuscript to make it read-able, and a resubmission to the journal. See some comments below:

  1. Title: There should be an “an” before “AISI”.
  2. Throughout the manuscript: please check thoroughly the usage of “a” and “the” before nouns. For example, in this sentence: “The control of the plastic properties can be carried out, for example, by using a predeformation loadings of material in the desired directions.” It should be “The control of plastic properties of a material can be carried out, for example, by using predeformation loading in desired directions.” In addition, please provide citations for this claim.
  3. Intro: “Typically, only certain grain orientations are preferred, and then such a material is characterized by a specific crystallographic texture. ” Can authors be specific and provide examples and citations for this?
  4. Intro: “For completeness, it was reference several models for plastically anisotropic metals” Please correct this sentence.
  5. Intro: “They have been designed 21 test protocols. ” Please correct this sentence. 
  6. Intro: “They have been designed 21 test protocols. Several samples are subjected to the proportional (baseline) loading and are brought to failure. These samples are axially deformed in the rolling direction, the transverse direction, and at 45 degrees to the rolling direction. Other samples are pre-deformed, unloaded, and then, reloaded in one of three selected directions.” Should these appear in the experimental section?
  7. Intro: “At each stage of the deformation, it is calculated the rates of the axial and lateral plastic strains. It is also introduced, the Huber-Mises plastic flow mechanism with added in terms designed to capture the plastic anisotropy. Under the assumption of plastic incompressibility, were shown that our two-stage tests in tension are sufficient for a complete calibration of the plastic flow mechanisms” Please correct the sentences.
  8. Materials and Methods: Figure 2 has no scale bars!

Author Response

Thank you again for your response concerning our manuscript entitled “Effect of initial predeformation on the plastic properties of rolled sheets of AISI 304L austenitic steel”. I have revised the manuscript according to reviewers’ comments. In the text marked in red made all the changes suggested by the reviewers. In addition, the structure of the article was restructured by including information about the experimental studies in the appendix. The table below contains the answers to reviewers suggestions. I hope the changes are sufficient to meet the standards of Materials both in terms of English language and manuscript content. Thank you for your time.

The introduction of the paper was modified, literature references were added, and examples for specific statements were provided.

Unclear sentences have been clarified.

The structure of the article was rewritten, including placing the experimental section in an appendix.

Added a scale to microscopic images of the material being examined.

Reviewer 4 Report

The article is devoted to the study of the anisotropy of properties in AISI 304L steel subjected to deformation and tension
Undoubtedly, the results presented by the authors are of high scientific novelty and practical significance, and are also promising for practical research. In general, the presented results of the study can be accepted for publication after the authors provide answers to all the questions raised by the reviewer during the reading of the article.

1. In the abstract, the authors need to more clearly state the purpose and relevance of this work.
2. Authors should change the presentation of the results, some of them can be combined together or put into an appendix.
3. The authors should explain why, for samples No. 1 and No. 2, the formation of a crack does not occur in the center of the sample.
4. The authors should explain what causes the formation of grains in the form of elongated rhombuses or rectangles after heating.
5. The authors should explain exactly how the trajectory and direction of deformation affect the mechanisms of plastic flow, as well as the influence of dislocation density on these effects.
6. Conclusion requires significant revision.

Author Response

Thank you again for your response concerning our manuscript entitled “Effect of initial predeformation on the plastic properties of rolled sheets of AISI 304L austenitic steel”. I have revised the manuscript according to reviewers’ comments. In the text marked in red made all the changes suggested by the reviewers. In addition, the structure of the article was restructured by including information about the experimental studies in the appendix. The table below contains the answers to reviewers suggestions. I hope the changes are sufficient to meet the standards of Materials both in terms of English language and manuscript content. Thank you for your time.

The abstract has been rewritten.

The structure of the article was rewritten, including placing the experimental section in an appendix.

Predeformation of type A specimens was induced up to 30%. Thermographic tests were to confirm that up to this range in the work section of the sample there is a uniform distribution of deformations, which is shown in Fig. 5. Fracture of samples A1 and A2 (cut in line and transverse to the rolling direction) occurred at strains much larger than those assumed in the tests and outside the middle zone of the sample. This could be due to the presence of inhomogeneities in the material in the form of inclusions and voids.

The appendix explains the reasons for the change in grain geometry in the structure of the annealed material.

In Chapter 3, an attempt is made to explain the factors affecting the plastic flow mechanisms of the material.

The summary has been rewritten.

Round 2

Reviewer 1 Report

The manuscript even after the revision, I feel that the report is not providing significant impact  in terms of scientific point of view. The work revolves around small scientific information and also the experimental section is not properly designed to cover the important aspects in this view. Therefore, I must reject the work.   

Author Response

I thank the reviewer for his time.
I believe that the approach to modeling the anisotropy of plastic properties of a material presented in this paper, due to its innovative solution from a scientific point of view, contributes to the development of the discipline of materials engineering. The presented method has an iplementation potential.

Reviewer 2 Report

Dear Szusta and Zubelewicz,

After carefully reading the author's revised manuscript entitled "Effect of initial predeformation on the plastic properties of rolled sheets of AISI 304L austenitic steel" (materials-1688874) by Szusta and Zubelewicz, The authors have been made significant improvements in the revised manuscript. Also, all of the issue in my review report have been addressed precisely.

With my pleasure, I recommend the manuscript should be accepted for publication on Materials.

Best regards,

The Reviewer

Author Response

Thank you for your insightful analysis of the developed publication and your valuable guidance.

Reviewer 3 Report

The revisions reach the satisfactory for publication on Materials.

Author Response

(The authors gave the same response as above.)
